# Does Multisession Cathodal Transcranial Direct Current Stimulation of the Left Dorsolateral Prefrontal Cortex Prime the Effects of Cognitive Behavioral Therapy on Fear of Pain, Fear of Movement, and Disability in Patients with Nonspecific Low Back Pain? A Randomized Clinical Trial Study

**DOI:** 10.3390/brainsci13101381

**Published:** 2023-09-28

**Authors:** Fatemeh Ehsani, Mohaddeseh Sadat Hafez Yousefi, Abbas Jafarzadeh, Maryam Zoghi, Shapour Jaberzadeh

**Affiliations:** 1Neuromuscular Rehabilitation Research Centre, Semnan University of Medical Sciences, Semnan 3514799442, Iran; fatemehehsani59@yahoo.com (F.E.); jafarzadehabbas8005@yahoo.com (A.J.); 2Department of Physiotherapy, School of Rehabilitation, Tehran University of Medical Sciences, Tehran 5166614711, Iran; 3Discipline of Physiotherapy, Institute of Health and Wellbeing, Federation University Victoria, Ballarat, VIC 3350, Australia; m.zoghi@federation.edu.au; 4Department of Physiotherapy, Faculty of Medicine, Nursing and Health Sciences, Monash University, Melbourne, VIC 3800, Australia; shapour.jaberzadeh@monash.edu

**Keywords:** low back pain, cognition, transcranial direct electrical stimulation, left dorsolateral prefrontal cortex, fear, disability

## Abstract

Many studies have shown that low back pain (LBP) is associated with psychosomatic symptoms which may lead to brain changes. This study aimed to investigate the effect of the concurrent application of cognitive behavioral therapy (CBT) and transcranial direct electrical stimulation (tDCS) over the left dorsolateral prefrontal cortex (DLPFC) on fear of pain, fear of movement, and disability in patients with nonspecific LBP. This study was performed on 45 LBP patients (23 women, 22 men; mean age 33.00 ± 1.77 years) in three groups: experimental (2 mA cathodal tDCS (c-tDCS)), sham (c-tDCS turned off after 30 s), and control (only received CBT). In all groups, CBT was conducted for 20 min per session, with two sessions per week for four weeks. Fear of pain, fear of movement, and disability were evaluated using questionnaires at baseline, immediately after, and one month after completion of interventions. Results indicated that all three different types of intervention could significantly reduce fear and disability immediately after intervention (*p* > 0.05). However, improvement in the experimental group was significantly higher than in the other groups immediately after and at the one-month follow-up after interventions (*p* < 0.05). DLPFC c-tDCS can prime the immediate effects of CBT and also the lasting effects on the reduction in the fear of pain, fear of movement, and disability in LBP patients.

## 1. Introduction

Low back pain (LBP) is one of the most common disabling musculoskeletal disorders, which involves about 70 to 85% of individuals in different countries [1,2,3]. Evidence indicates that chronic LBP could affect the function and structure of some parts of the brain in addition to the function and structure of the lumbar spine [4]. These changes could in turn lead to abnormal changes in the processing of sensorimotor signals and psychological processes in the brain [5,6,7]. After experiencing chronic pain, several psychosocial factors, such as the attitude and beliefs of the patients about their pain, are changed. These changes may lead to increases in fear of pain and movement, stress, and depression [8,9].

Evidence indicates that patients with chronic LBP who experience high levels of pain-related anxiety have significantly higher levels of disability [10,11,12]. In addition, a high level of pain-related anxiety could reduce the function of back stabilizer muscles, which may hurt the routine treatment in chronic LBP patients [13,14]. Therefore, control of pain-related anxiety in these patients is an important step in the rehabilitation of these patients. Cognitive behavioral therapy (CBT) is one of the effective interventions for the treatment of pain-related anxiety and beliefs following chronic pain [15,16]. Aneis et al. (2021) indicated that the pain severity, disability, and pain-related anxiety were significantly reduced after CBT treatments condcuted up to three times per week for six weeks in patients with nonspecific chronic LBP [17]. However, a recent systematic review indicated that while CBT can reduce pain and disability immediately after treatment, its lasting effects in the follow-up period are not clear [18].

Transcranial direct current stimulation (tDCS), is a noninvasive neuromodulatory technique that recently gained popularity for various purposes [19,20]. In addition, a study in 2020 showed that tDCS is effective in reducing pain [21]. The literature indicates that tDCS may induce sustained neuroplastic changes in the brain in line with changes during the long-term potentiation (LTP) mechanism [22,23]. The significant effect of tDCS on the management of chronic pain is documented in patients with trigeminal neuralgia, poststroke pain syndrome, fibromyalgia, and LBP [20]. In parallel, several studies showed the significant effects of tDCS on the treatment of depression and fear of movement in patients with major depression and other psychological disorders [24,25]. Asthana et al. (2013) investigated the effects of cathodal tDCS (c-tDCS) and anodal tDCS (a-tDCS) over the dorsolateral prefrontal cortex (DLPFC) on the experimental memory of fear in a healthy individual and concluded that c-tDCS over the DLPFC could induce a significant reduction in the memory of fear, while a-tDCS of the DLPFC failed to do so [26]. Another study evaluated the effect of concurrent CBT and a-tDCS over the primary motor cortex (M1) on the reduction in pain and disability in patients with nonspecific chronic LBP [15]. The study concluded that M1 a-tDCS did not have any significant effect on the reduction in pain and disability in CLBP patients [15]. In other words, the studies showed that the c-tDCS method on the DLPFC can inhibit the generation of fear in the memory and, unlike the anodal method, it can increase fear memories [27]. It was also reported that the prefrontal cortex (DLPFC) region is involved in the formation of fear in the memory. On the other hand, the evidence showed that the inhibition of the left DLPFC causes adjustments in the level of anxiety and fear [28,29].

Therefore, in keeping with the reviewed studies, this study aimed to investigate the short-term and lasting effects of concurrent CBT and c-tDCS over the DLPFC on pain, pain-related anxiety, and fear of movements in patients with LBP with high pain-related anxiety. We hypothesized the following:Multiple sessions of a CBT program or sham tDCS with CBT reduces pain-related anxiety, fear of movement, and disability in LBP patients with high pain-related anxiety.Multiple sessions of concurrent CBT with c-tDCS over the left DLPFC will be more effective for the reduction in the pain-related anxiety, fear of movement, and disability in LBP patients with high pain-related anxiety compared to CBT alone or concurrent CBT and sham c-tDCS of the left DLPFC.Multiple sessions of concurrent CBT with c-tDCS over the left DLPFC will have a more lasting effect on the reduction in the pain-related anxiety, fear of movement, and disability in LBP patients with high pain-related anxiety compared to CBT alone or concurrent CBT and sham c-tDCS of the left DLPFC.

## 2. Method and Materials

### 2.1. Participants

Fifty-six patients with nonspecific LBP were assessed against the inclusion and exclusion criteria. Overall, 8 patients were excluded and 45 CLBP patients (23 female, 22 male) with high pain-related anxiety, aged between 18 and 45 (33.00 ± 1.74), participated in this study (Figure 1).

Inclusion criteria were as follows: (1) suffering from LBP for more than six weeks or having recurrent LBP with at least three episodes lasting more than one week (subacute LBP) [30,31]; (2) a pain score between 1 and 3 out of 10 on a visual analog scale (VAS) on the testing day [26]; (3) a score above 30 on the Pain Anxiety Symptoms Scale (PASS) [32].

Exclusion criteria included reporting any history of neurological diseases, such as Parkinson’s disease, Alzheimer’s, and cerebellar disorders; reporting any history of psychological illnesses; the presence of any signs of radiculopathy or root lumbar spinal cord involvement; structural deformities in the spine, such as scoliosis, kyphosis, or lordosis; and any abnormalities in the vestibular system [33,34,35].

All patients provided written informed consent before inclusion in this study. This study conforms to the consort checklist criteria. Eligible individuals were randomly based on computer coding assigned into three groups: (1) concurrent left DLPFC c-tDCS and CBT (experimental group); (2) concurrent sham c-tDCS and CBT (sham stimulation group); (3) CBT alone (control group).

### 2.2. Study Design

This is a randomized, parallel, double-blind sham-controlled study. From a pool of 56 LBP participants who were referred from neurological clinics by a neurologist, 48 LBP participants with high pain-related anxiety met the inclusion and exclusion criteria. The participants were randomly assigned to three groups following a computer-generated list of random numbers (n = 16 in each group, Figure 1). The number of patients was estimated based on Cohen’s table. This sample size allows detection of the effect of DLPFC c-tDCS on the outcome measures in this study with the power of 85 and 95% confidence intervals (CI). All participants in the first two groups received concurrent c-tDCS or sham c-tDCS and CBT. Participants in the control group received only CBT. Fear of pain, fear of movement, and disability were evaluated using the Pain Anxiety Symptoms Scale (PASS), Tampa Scale for Kinesiophobia (TSK), and Roland–Morris questionnaire (RMDQ), respectively, before, immediately after, and one month after the interventions. Finally, 45 participants completed the whole program of study (Figure 1). Throughout this study, one of the researchers was responsible for the application of the interventions in each group, and the second researcher, blinded to the treatment groups, was responsible for the assessment of the outcome measures. All participants were also blinded to the application of sham or active c-tDCS.

This study was approved by the Human Ethics Committee of the Semnan University of Medical Sciences, Semnan, Iran (IR.SEMUMS.REC.1396.162) and was performed following the ethical standards laid down by the Declaration of Helsinki. This study was registered as a clinical trial on the Iranian Registry of Clinical Trials (IRCT20150302021294N6).

### 2.3. Transcranial Direct Current Stimulation

In both the experimental and sham stimulation groups, 2 mA c-tDCS (ActivaDose^®^ II, ActivaTeK™Inc., Gilroy, CA, USA) was applied for 20 min using large electrodes (5 × 7 cm) [36,37]. In the first and the last 10 s of stimulation, the current was gradually ramped up or down [38,39] to avoid any sudden changes in the induced sensations [40]. In the experimental group, the active electrode (cathode) and the returning electrode (anode) were placed over the left DLPFC (F3, 10–20 international encephalography system) and the right supraorbital region, respectively [41] (Figure 2).

In the sham c-tDCS group, the “Fade-in Short stimulation Fade-out (FiSsFo) approach” was used [37,41], which is a reliable method for maintaining the blinding integrity and thus creating the assumed initial cutaneous sensations. In this group, the electrode montage was identical to the active stimulation, and the stimulation was slowly turned off after 30 s [42].

### 2.4. Cognitive Behavioral Therapy (CBT)

All participants received 8 sessions of CBT administered by a psychologist expert in this field of work [43]. Eight different topics were covered in the 8 treatment sessions of CBT, including identification of participants’ beliefs about pain and pain treatment as well as reconceptualization of the pain experience (session 1); education about the various theories of pain and relaxation techniques such as diaphragmatic breathing, progressive muscle relaxation, and visual imagery (sessions 2–4); education regarding the importance of scheduling pleasant activities and thought, a time-based pacing technique in which individuals take breaks based on time or amount of work accomplished rather than pain level (sessions 5–6); cognitive restructuring to help participants learn to identify maladaptive thoughts and beliefs related to pain and substitute more adaptive ones (session 7); and education of additional training in techniques related to anger management and sleep hygiene (session 8).

In addition, participants were given homework assignments at the end of each session to practice the various techniques presented during the previous sessions. The therapist also collaborated with each patient to generate specific intersession goals (e.g., daily walks for 20 min) after each session. From the second session, a review of the homework assignments and intersession goal accomplishment; problem-solving discussions; a brief review of materials covered in the previous session; the presentation and practice of new skills; and the collaborative establishment of new homework/goals for the next session were performed by the therapist in each session [44].

### 2.5. Outcome Measures

Persian translation of the TSK, as a valid and reliable scale [45,46], was used to assess the fear avoidance belief in the movement in patients. The TSK is a 17-item scale with a four-point Likert score for each item from 1 (strongly disagree) to 4 (strongly agree). The minimum–maximum score for the kinesiophobia in the TSK is 17–68. Higher scores showed higher kinesiophobia or fear of movement [47,48].

Persian translation of the PASS, as a valid and reliable scale [32], was used to investigate the pain-related anxiety rate. This questionnaire includes 14 subscales of pain-related cognitive anxiety, escape, pain-related avoidance, pain-related fear, and pain-related physiologic anxiety symptoms. The short form of this scale has 20 items with a 5-point score for each item from 0 (never) to 5 (always). The total score is 100, and those who obtained a score higher than 30 are categorized as patients with high pain-related anxiety [24].

To measure the disability level, the Roland–Morris questionnaire, which is a valid and reliable questionnaire [49,50], was used. This questionnaire has 24 statements that describe the possibility of limiting the probable activities. Scores range from 0 (indicating no disability) to 24 (indicating severe disability) [49,50].

## 3. Results

Table 1 shows the demographic details and baseline data for each group. There were no significant differences in variables of age, gender, and PASS, TSK, and RMDQ scores among the groups (*p* > 0.05).

RM-ANOVA shows a significant main effect of “Time” and also the “Group” × “Time” interaction effect for PASS, TSK, and RMDQ scores (*p* < 0.01) (Table 2). Accordingly, post hoc tests were performed using Bonferroni correction (Table 3). The findings indicated a significant reduction in the PASS, TSK, and RMDQ scores in the experimental group immediately and one month after the intervention (*p* < 0.001, Table 3, Figure 3, Figure 4 and Figure 5). In addition, a significant reduction in the PASS, TSK, and RMDQ scores was shown in the sham and control groups immediately after the intervention (*p* < 0.001, Table 3, Figure 3, Figure 4 and Figure 5), while this reduction did not remain one month after the intervention (*p* > 0.05, Figure 3, Figure 4 and Figure 5). Moreover, although PASS, TSK, and RMDQ scores decreased in the sham and control groups immediately after the intervention, the reduction in PASS, TSK, and RMDQ scores in the experimental group was significantly more than in the sham and control groups immediately and one month after completion of the intervention (Figure 3, Figure 4 and Figure 5, *p* < 0.01).

### Safety and Side Effects of c-tDCS

Table 4 shows the side effects (mean ± SEM) under the anode and cathode that were reported by all participants. Itching was the side effect reported by the majority of participants. None of the participants reported any burning sensation or pain during or after c-tDCS. This study indicated that DLPFC c-tDCS intervention was tolerated very well with minimal adverse side effects by all participants. The presence and severity of the possible side effects were determined by a questionnaire. This questionnaire included a rating scale and numeric analog scale (NAS) (e.g., 0 = no tingling to 10 = worst tingling imaginable).

## 4. Discussion

In this study, for the first time, the short-term and lasting priming effects of multisession concurrent c-tDCS of the left DLPFC and CBT on pain-related anxiety, fear of movement, and disability level of LBP patients with high pain-related anxiety were studied. The findings in the current study indicated that there was a significant decrease in the PASS, TSK, and Roland–Morris questionnaire scores immediately after the interventions in sham and intervention groups. Also, it showed that the combination of c-tDCS and CBT programs was more effective in the reduction in pain-related anxiety, fear of movement, and disability compared to CBT programs alone. The results of the present study showed that the significant improvement in kinesiophobia, pain-related anxiety, and disability after treatment remained for one month postintervention.

In this study, for the first time, the short-term and lasting priming effects of multisession concurrent c-tDCS of the left DLPFC and CBT on pain-related anxiety, fear of movement, and disability level of LBP patients with high pain-related anxiety were studied. We hypothesized that CBT alone could reduce pain-related anxiety, fear of movement, and disability. The findings in the current study supported this hypothesis and indicated that there was a significant decrease in the PASS, TSK, and Roland–Morris questionnaire scores immediately after the interventions in all groups (*p* < 0.001). This finding is in line with the findings of other studies that concluded that CBT can significantly reduce the pain severity, fear of pain, and disability and also improve the quality of life in patients with chronic LBP [44,51]. In addition, a study on elderly patients with chronic LBP indicated that pain severity and disability were significantly reduced up to two weeks after a 10-session course of CBT [44]. Monticone et al. (2015) also investigated the effects of exercise and CBT on fear of pain, disability, and pain severity in patients with nonspecific chronic LBP [44]. The study indicated that CBT along with exercise as compared to exercise alone can significantly improve the fear of pain, disability, pain severity, and quality of life in patients with nonspecific chronic LBP [44].

The findings in the current study also supported the second hypothesis with the fact that the combination of c-tDCS and the CBT program was more effective in the reduction in pain-related anxiety, fear of movement, and disability compared to the CBT program alone (*p* < 0.001). In this regard, several review studies concluded that although CBT, as compared to the other interventions, is an effective intervention in the treatment of chronic pain syndromes such as chronic LBP, its effect on fear, anxiety, and pain is not long-lasting [16,47,48,52]. A recent systematic review in 2022 indicated that CBT can reduce pain and disability immediately after treatment, but its lasting effects in the follow-up period are not clear [18]. However, the results of another review study conducted by Richmond et al. (2015) showed that CBT could have positive short-term and long-term effects on pain and disability in acute LBP patients [46]. It seems that the reason behind this discrepancy lies in the differences between the stages of LBP in these studies. Richmond et al. (2015) included the studies that assessed the effect of CBT on acute LBP, while other studies, like the current study, assessed the effect of CBT on chronic LBP patients [52]. Indeed, unlike acute pain, chronic pain can lead to functional and structural changes in different areas of the brain. Chronic low back pain (CLBP) has a more complex nature compared with acute episodes, since cognitive, emotional, behavioral, and social factors directly affect the CLBP experience [53].

The results of the present study showed that the significant improvement in kinesiophobia, pain-related anxiety, and disability after treatment remained for one month postintervention (*p* < 0.001), while in CBT alone and CBT with sham c-tDCS groups, the results were not significant after one month (*p* > 0.05). Middlkoop et al. (2010), in their systematic review study, stated that the tDCS technique along with CBT can modulate the central nervous system and so increase the effects of CBT in nonspecific chronic LBP patients [54]. There is evidence that tDCS can lead to functional and structural effects on the cortex and facilitate neural plasticity by a mechanism similar to long-term potential (LPT) [22,23,47]. In addition, some studies reported the positive and significant effects of tDCS on the pain, disability, fear, and anxiety in patients with chronic pain such as chronic LBP [28,54,55]. Results of the Antal study also show that c-tDCS has significant short-term and long-term effects on reducing chronic pain and modulating neuronal activity [47]. Furthermore, the Mariano study found that 10 sessions of 2 mA c-tDCS on the left dorsal anterior cingulate cortex (dACC) could modulate CLBP’s affective symptoms such as pain intensity, acceptance, interference, disability, and anxiety, plus general anxiety and depression immediately and 6 weeks after intervention [56]. However, Kerstin Leudtke et al. (2015) examined the effect of M1 a-tDCS alone and along with CBT on reducing the pain and disability in nonspecific chronic LBP patients and showed that none of them had any effect on reducing pain and disability [15]. There is evidence that the prefrontal cortex is one of the important brain areas involved in the formation of fear and pain-related anxiety following chronic pain [26,29]. Therefore, it seems that this area has a key role in modulating the neural activity and memory of fear during chronic pain. Furthermore, this study showed that DLPFC c-tDCS could decrease the fear while a-tDCS could increase it in healthy humans [33]. The results of the current study also showed that concurrent CBT and left DLPFC c-tDCS as compared to CBT with sham DLPFC c-tDCS or CBT alone can reduce pain-related anxiety, kinesiophobia, and disability in LBP patients with high pain-related anxiety.

The findings of this study have to be seen in light of some limitations. One of the limitations of this study was the age of the participants, who were young adults. Therefore, the results cannot be generalized to middle-aged or older CLBP adults. Therefore, we recommend conducting further studies to investigate the effect of DLPFC c-tDCS on pain-related anxiety, fear of movement, and disability levels in middle-aged or older patients with LBP. In addition, the current study considered only a one-month follow-up. Longer follow-up assessments are also recommended in further studies. Moreover, the main outcome measures for anxiety, fear, and disability were not in quantity in the current study. Conducting future studies is suggested to quantify the outcome measures.

## 5. Conclusions

The findings in this study provided evidence for the priming effects of multiple-session c-tDCS over the left DLPFC on the effects of CBT on pain-related anxiety, fear of movement, and disability in LBP patients with high pain-related anxiety. The findings in the current study indicated significant decreases in PASS, TSK, and Roland–Morris questionnaire scores immediately after the interventions in all groups. However, these immediate effects were more in the c-tDCS and CBT group compared to the CBT alone and the sham groups. The findings in this study also confirmed the lasting effects of c-tDCS along with CBT in LBP patients with high pain-related anxiety.

## Figures and Tables

**Figure 1 brainsci-13-01381-f001:**
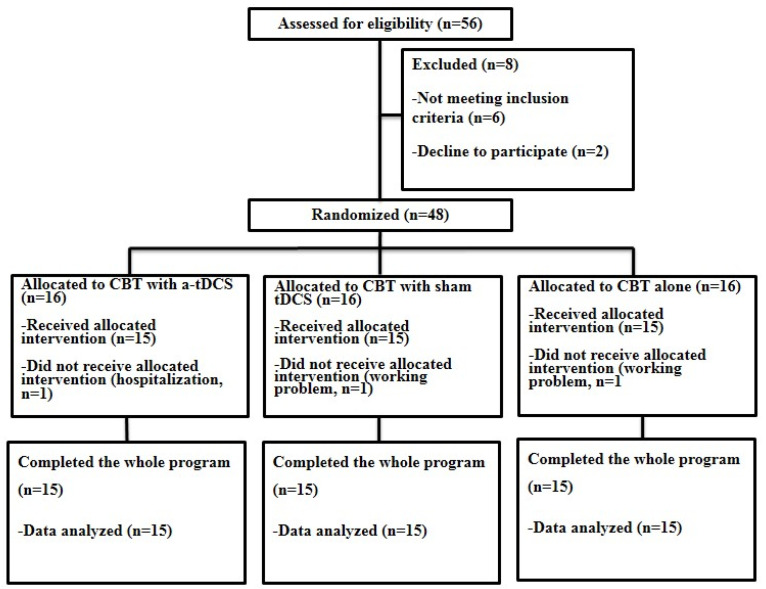
Flow diagram of participants’ eligibility assessment.

**Figure 2 brainsci-13-01381-f002:**
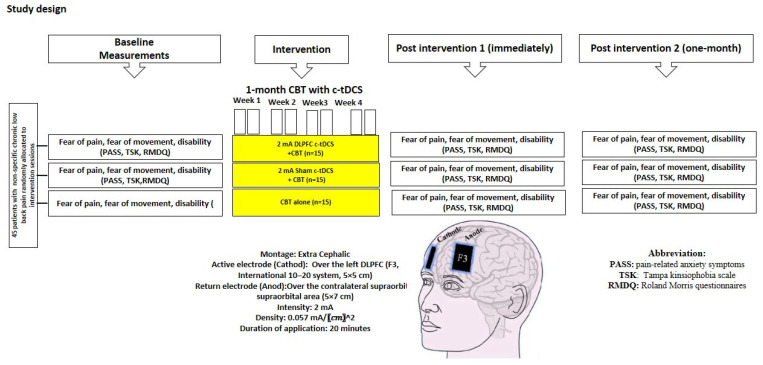
Experimental design: 8 sessions of interventions in three groups. PASS, TSK, and RMDQ questionnaires were used for assessment before, immediately after, and 1 month after intervention.

**Figure 3 brainsci-13-01381-f003:**
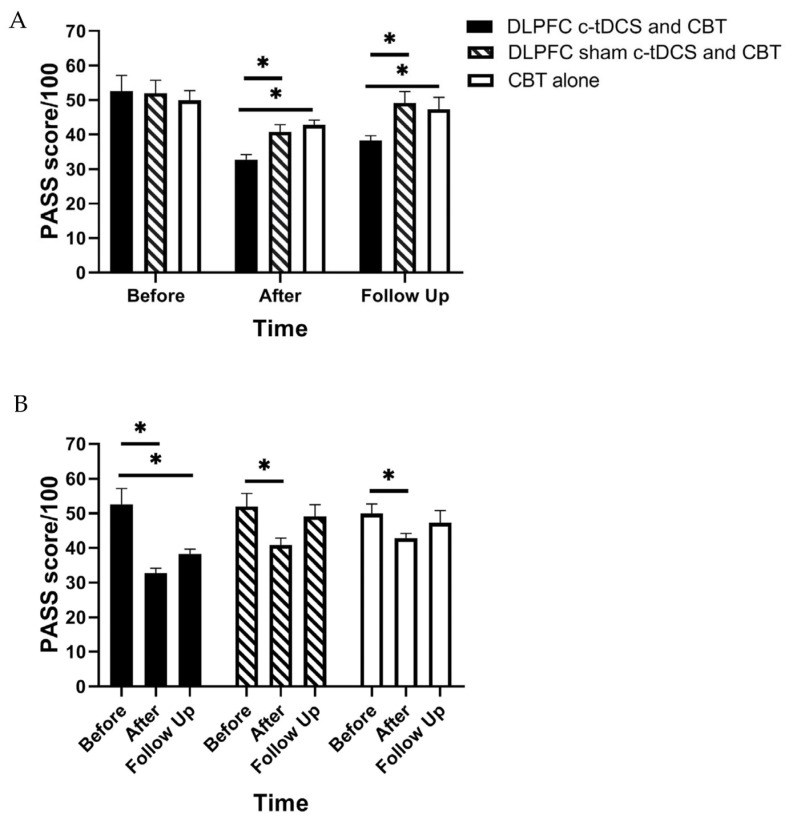
(**A**) The comparison of the Pain Anxiety Symptoms Scale (PASS) scores (mean ± SEM) before, immediately after, and one month after the intervention in DLPFC c-tDCS concurrent with CBT, sham DLPFC c-tDCS concurrent with CBT, and CBT alone groups; * indicates the significant differences in PASS score after intervention between groups. (**B**) The comparison of the Pain Anxiety Symptoms Scale (PASS) scores (mean ± SEM) before, immediately after, and one month after the intervention in each group; * indicates the significant differences in PASS score after intervention rather than baseline in each group.

**Figure 4 brainsci-13-01381-f004:**
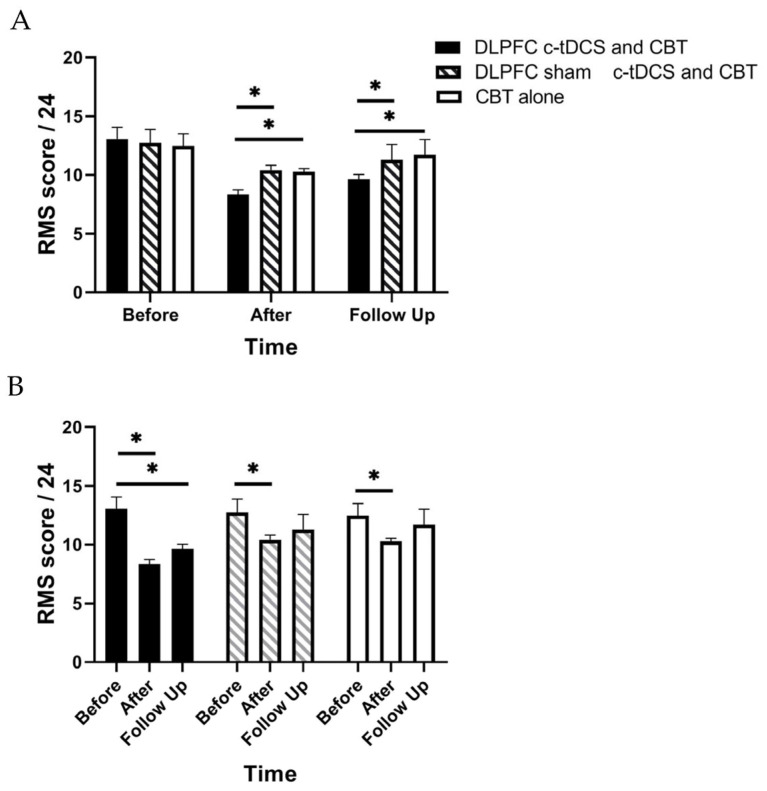
(**A**) The comparison of the Roland–Morris Scale (RMS) scores (mean ± SEM) before, immediately after, and one month after the intervention in DLPFC c-tDCS concurrent with CBT, sham DLPFC c-tDCS concurrent with CBT, and CBT alone groups; * indicates the significant differences in RMS score after intervention between groups. (**B**) The comparison of the Roland–Morris Scale (RMS) scores (mean ± SEM) before, immediately after, and one month after the intervention in each group; * indicates the significant differences in RMS score after intervention rather than baseline in each group.

**Figure 5 brainsci-13-01381-f005:**
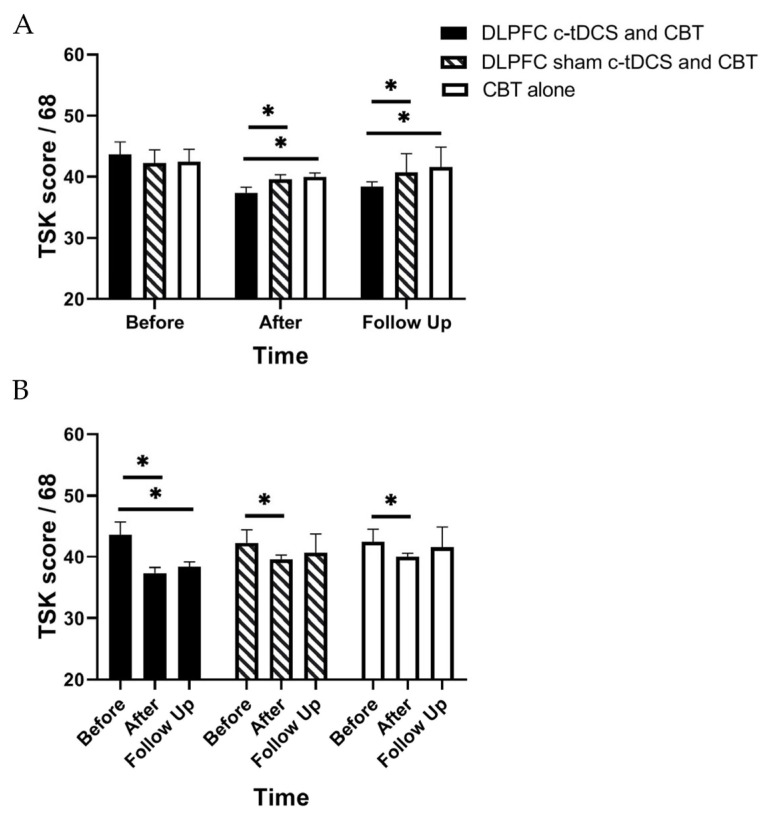
(**A**) The comparison of the Tampa Scale for Kinesiophobia (TSK) scores (mean ± SEM) before, immediately after, and one month after the intervention in DLPFC c-tDCS concurrent with CBT, sham DLPFC c-tDCS concurrent with CBT, and CBT alone groups; * indicates the significant differences in TSK score after intervention between groups. (**B**) The comparison of the Tampa Scale for Kinesiophobia (TSK) scores (mean ± SEM) before, immediately after, and one month after the intervention in each group; * indicates the significant differences in TSK score after intervention rather than baseline in each group.

**Table 1 brainsci-13-01381-t001:** Demographic data and baseline values for the participants in experimental and sham groups (mean ± SEM). PASS (Pain Anxiety Symptoms Scale), TSK (Tampa Scale for Kinesiophobia), RMDQ (Roland–Morris Disability Questionnaire), DLPFC (dorsolateral prefrontal cortex), CBT (cognitive behavioral therapy).

Variables	DLPFC tDCS+ CBT		Sham tDCS+ CBT		Control		*p* Value
	SEM	Mean	SEM	Mean	SEM	Mean	
Age	1.60	33.00	1.82	33.00	1.77	33.00	0.93
Gender (male/female)	-	7/8	-	7/8	-	8/7	0.89
PASS	4.53	52.60	3.82	51.93	2.78	49.93	0.87
TSK	2.05	43.66	2.14	42.26	2.04	42.46	0.87
RMDQ	1.37	13.06	1.15	12.73	1.04	12.46	0.93

**Table 2 brainsci-13-01381-t002:** ANOVA results for the effects of c-tDCS on fear of pain, fear of movement, and disability. PASS (Pain Anxiety Symptoms Scale), TSK (Tampa Scale for Kinesiophobia), RMDQ (Roland–Morris Disability Questionnaire), and CBT (cognitive behavioral therapy).

Variables	Effect		DF	F	*p* Value
PASS	Main	Group	2	0.24	0.90
		Time	2	26.11	* <0.001
	Interaction	Group × Time	4	4.47	* 0.001
TSK	Main	Group	2	0.01	0.990
		Time	2	7.79	* <0.001
	Interaction	Group × Time	4	9.78	* <0.001
RMDQ	Main	Group	2	0.10	0.990
		Time	2	10.26	* <0.00
	Interaction	Group × Time	4	9.78	* <0.001

**Table 3 brainsci-13-01381-t003:** Post hoc pair-wise comparison of PASS (Pain Anxiety Symptoms Scale), TSK (Tampa Scale for Kinesiophobia), and RMDQ (Roland–Morris Disability Questionnaire) scores before and after intervention in each group.

Group	Variables	Time Assessment		Mean Difference (95% CI)	*p* Value
DLPFC c-tDCS with CBT	PASS	T_1_	T2	19.93 (11.95–27.91)	* <0.001
			T3	23.13 (10.11–36.14)	* 0.004
	TSK		T2	6.33 (2.62–10.05)	* 0.003
			T3	8.12 (2.01–14.24)	* 0.016
	RMDQ		T2	4.73 (2.89–6.57)	* <0.001
			T3	5.12 (1.98–8.27)	0.006
Sham DLPFC c-tDCS with CBT	PASS	T_1_	T2	11.13 (6.83–15.43)	* <0.001
			T3	7.71 (−17.23–1.80)	0.095
	TSK		T2	2.67 (1.74−3.59)	* <0.001
			T3	1.57 (0.02–3.16)	0.052
	RMDQ		T2	2.33 (1.88–2.78)	* <0.001
			T3	1.86 (−0.24–3.95)	0.073
CBT alone	PASS	T_1_	T2	8.46 (6.41–10.51)	* <0.001
			T3	5.14 (−12.60–2.32)	0.143
	TSK		T2	2.46 (1.80–3.12)	* <0.001
			T3	1.57 (−1.71–4.86)	0.286
	RMDQ		T2	2.20 (1.53–2.87)	* <0.001
			T3	1.28 (−1.02–3.59)	0.222

**Table 4 brainsci-13-01381-t004:** Evaluation of side effects during tDCS intervention in both groups (numeric sensation scores).

		Cathodal Electrode		Anodal Electrode	
		DLPFC tDCS	Sham tDCS	DLPFC tDCS	Sham tDCS
Tingling sensation	Beginning	0.65 ± 0.04	0.52 ± 0.14	4.14 ± 0.18	2.54 ± 0.15
	Middle	0.75 ± 0.15	0.49 ± 0.19	5.19 ± 0.13	1.36 ± 0.14
	End	0.67 ± 0.12	0.51 ± 0.08	2.28 ± 0.25	0.77 ± 0.15
Itching sensation	Beginning	0.87 ± 0.23	0.79 ± 0.11	3.36 ± 0.38	1.03 ± 0.25
	Middle	0.71 ± 0.17	0.47 ± 0.08	2.53 ± 0.27	0.95 ± 0.19
	End	0.73 ± 0.13	0.21 ± 0.04	1.73 ± 0.41	0.74 ± 0.08
Burning sensation	Beginning	-	-	-	-
	Middle	-	-	-	-
	End	-	-	-	-
Pain	Beginning	-	-	-	-
	Middle	-	-	-	-
	End	-	-	-	-
Headache	Beginning	-	-	-	-
	Middle	-	-	-	-
	End	-	-	-	-
Not tolerated others	Beginning	-	-	-	-
	Middle	-	-	-	-
	End	-	-	-	-

## Data Availability

The data presented in this study are available on request from the corresponding author.

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
