# Peer review of "Does Multisession Cathodal Transcranial Direct Current Stimulation of the Left Dorsolateral Prefrontal Cortex Prime the Effects of Cognitive Behavioral Therapy on Fear of Pain, Fear of Movement, and Disability in Patients with Nonspecific Low Back Pain? A Randomized Clinical Trial Study"

_brainsci, 2023, doi:10.3390/brainsci13101381_

Round 1

Reviewer 1 Report

Dear Authors,

It is my pleasure to review your study. The study is interesting. It raises the important issue of treating chronic pain but I have a lot of doubts.

General information:

-in the abstract, basic information about participants should be added / the results should be more precise,

-first of all, references should be corrected. References are too old. 

-references should be in square brackets in accordance with the guidelines of the journal.

Introduction:

-in line 48: "Ried et al 48 (2003)" - this is too old, cited references should be newer.

-in line 50: "However, a recent systematic review indicated that CBT can only have short-term effects on reduction of pain-related anxiety" - recent? This references is from 2011 - it's still old. In my opinion it should be corrected. 

-in line 54: recently may change to last 10 years ?, cited references are form 2014, 2015, 2013.

M&M:

-in the inclusion criteria: "Suffering from LBP for more than six weeks" but authores used termin chronic. Chronic pain lasts 12 or more weeks. 

-the inclusion and exclusion criteria are hard to read, it should be corrected.

-it should be clarified who examined patients with LBP? What diagnostic imaging was used? etc.

-the division into groups should be described in more detail.

-please describe the study participants in more detail.

-"This study was approved by the Human Ethics Committee" please add the consent number.

-was the sample size counted?

Results:

-figure 1, 2, 3, 4 and 5 should be described.

-abbreviations for table 1, 2, 3 and 4 should be provided.

Discussion:

-discussion needs correction, more recent references should be quoted,

-it is necessary to refer more precisely to your results in the discussion, currently it is presented too generally.

English proofreading is required.

Author Response

Response to the comments of reviewer 1:

General information:

  • in the abstract, basic information about participants should be added / the results should be more precise

Response: Basic information about participants and results is added in the abstract in lines 18,19,25,26 on page 1.

  • -first of all, references should be corrected. References are too old

Response: New references (2019, 2022) are added in the introduction and discussion sections of the manuscript.

  • -references should be in square brackets to the guidelines of the journal.

Response: All references are placed in the brackets.

Introduction:

  • in line 48: "Ried et al 48 (2003)" - this is too old, cited references should be newer.

Response: This reference was deleted (lines 49-53) and new references are added in the whole manuscript.

  • in line 50: "However, a recent systematic review indicated that CBT can only have short-term effects on reduction of pain-related anxiety" - recent? This reference is from 2011 - it's still old. In my opinion, it should be corrected.
  • Response: The new reference is replaced in lines 52 and 53.

  • in line 54: recently may change to last 10 years ?, cited references are from 2014, 2015, and 2013: Response: The authorities are updated (line 54).

M&M:

  • in the inclusion criteria: "Suffering from LBP for more than six weeks" but the authors used the term chronic. Chronic pain lasts 12 or more weeks.

Response:  That is right. However, based on the results of some studies, which added in line 100, brain changes towards chronic pain occurred from the 6th week. Accordingly, LBP for more than six weeks is considered chronic pain in the current study.

  • the inclusion and exclusion criteria are hard to read, they should be corrected.

Response:  The paragraph related to inclusion and exclusion criteria is modified (lines 99-107).

  • it should be clarified who examined patients with LBP. What diagnostic imaging was used? etc. Response: It was clearly stated that the patients were first examined by a neurologist and referred to us (lines 113 and 114).
  • the division into groups should be described in more detail.

Response:  In lines 115 and 116, the grouping method is added with details.

  • please describe the study participants in more detail:

Response:  It was mentioned in more detail (lines 92, 93).

  • -"This study was approved by the Human Ethics Committee" Please add the consent number : Response: The consent number is added in line 130.

  • -was the sample size counted?

Response:  In line of 117, it is added that it was estimated based on Cohen's table.

Results:

  • -Figures 1, 2, 3, 4, and 5 should be described as:

Response:  All of the figures are described by adding the captions.

  • -abbreviations for tables 1, 2, 3, and 4 should be provided.

Response:  The abbreviations of all tables are added.

Discussion:

  • -The discussion needs correction, more recent references should be quoted:

Response:  According to the reviewer's comment, the discussion section is revised and also the new recent references are added (pages 11, 12).

  • -it is necessary to refer more precisely to your results in the discussion, currently, it is presented too generally:

Response:  According to the reviewer's comment, the whole discussion section is edited to present the results of the current study and the other studies more precisely (pages 11, 12).

Reviewer 2 Report

My comments are:

·       The authors cannot say long term because they measure the results at 1 month. That's short term.

·       The keywords are not well chosen. They must be appropriate words that reflect the theme of the paper.

·       The introduction should be longer. In the end, the authors must establish a justification and their hypothesis.

·       Lines 66-82: This text should be written differently.

·       All figures must have a caption.

·       The verb tense of the paper is present. Authors must write in the past tense

·       The tables must have the format required by the journal.

·       Line 250, 261, 267, 281: the refence is not correct. Review all paper.

·       The conclusion should be more concise and highlight the great discovery of the paper.

It's regular

Author Response

Response to the comments of reviewer 2:

  • The authors cannot say long-term because they measure the results at 1 month.

Response: According to the reviewer's comment, the word “long term” is changed to “lasting effect” in the whole manuscript.

  • The keywords are not well chosen. They must be appropriate words that reflect the theme of the paper.

Response: The keywords are changed on the first page.

  • The introduction should be longer. In the end, the authors must establish a justification and their hypothesis.

Response: According to the reviewer's comment, the introduction section is revised with more details to establish a justification. In addition, the hypotheses are added at the end of the introduction (page 2).

  • Lines 66-82: This text should be written differently.

Response: the sentences of these lines are edited.

  • All figures must have a caption.

Response: The caption is added for all figures.

  • The verb tense of the paper is present. Authors must write in the past tense.

Response: most of the verbs in the manuscript are corrected to present the past tense.

  • The tables must have the format required by the journal.

Response: All of the tables are changed according to the format of the journal.

  • Line 250, 261, 267, 281: the reference is not correct. Review all papers.

Response: All of the references of the manuscript, such as noted references in these lines, are checked again and corrected.

  • The conclusion should be more concise and highlight the great discovery of the paper.

Response: According to the reviewer's comment, the conclusion section is revised to show the results of the study more concisely and highlight (Page 13).

Round 2

Reviewer 1 Report

Dear Authors,

thank you for improving the article. However, there are still doubts.

General information:

  • in the abstract, basic information about participants should be added / the results should be more precise

Response: Basic information about participants and results is added in the abstract in lines 18,19,25,26 on page 1. That's not enough. Please add for example: (X women and Y men; mean age: ... ± ... years). The abstract is intended to be the essence of the entire manuscript. Must contain basic information.

  • -first of all, references should be corrected. References are too old

Response: New references (2019, 2022) are added in the introduction and discussion sections of the manuscript. Good, but I still think it is insufficient. Only 3 new articles have been added.

  • -references should be in square brackets to the guidelines of the journal.

Response: All references are placed in the brackets. OK.

Introduction:

  • in line 48: "Ried et al 48 (2003)" - this is too old, cited references should be newer.

Response: This reference was deleted (lines 49-53) and new references are added in the whole manuscript. OK.

  • in line 50: "However, a recent systematic review indicated that CBT can only have short-term effects on reduction of pain-related anxiety" - recent? This reference is from 2011 - it's still old. In my opinion, it should be corrected.
  • Response: The new reference is replaced in lines 52 and 53. OK.

  • in line 54: recently may change to last 10 years ?, cited references are from 2014, 2015, and 2013: Response: The authorities are updated (line 54). OK.

M&M:

  • in the inclusion criteria: "Suffering from LBP for more than six weeks" but the authors used the term chronic. Chronic pain lasts 12 or more weeks.

Response: That is right. However, based on the results of some studies, which added in line 100, brain changes towards chronic pain occurred from the 6th week. Accordingly, LBP for more than six weeks is considered chronic pain in the current study. 

I don't agree. Chronic pain is 12 weeks and over. Additionally, you cite the manuscript

"Medrano-Escalada Y, Plaza-Manzano G, Fernández-de-Las-Peñas C, Valera-Calero JA. Structural, functional, and 429 neurochemical cortical brain changes associated with chronic low back pain. Tomography 2022; 8: 2153-63."

where is the division:

"Clinically, it can be presented as acute pain episodes if <4 weeks of duration, subacute if 4–12 weeks of duration or chronic if >3 months of duration."

This absolutely needs to be improved throughout the article.

  • the inclusion and exclusion criteria are hard to read, they should be corrected.

Response: The paragraph related to inclusion and exclusion criteria is modified (lines 99-107). OK.

  • it should be clarified who examined patients with LBP. What diagnostic imaging was used? etc. Response: It was clearly stated that the patients were first examined by a neurologist and referred to us (lines 113 and 114). OK.
  • the division into groups should be described in more detail.

Response: In lines 115 and 116, the grouping method is added with details. OK.

  • please describe the study participants in more detail:

Response: It was mentioned in more detail (lines 92, 93). OK.

  • -"This study was approved by the Human Ethics Committee" Please add the consent number : Response: The consent number is added in line 130. OK.

  • -was the sample size counted?

Response: In line of 117, it is added that it was estimated based on Cohen's table. OK.

Results:

  • -Figures 1, 2, 3, 4, and 5 should be described as:

Response: All of the figures are described by adding the captions. OK.

  • -abbreviations for tables 1, 2, 3, and 4 should be provided.

Response: The abbreviations of all tables are added. OK.

Discussion:

  • -The discussion needs correction, more recent references should be quoted:

Response: According to the reviewer's comment, the discussion section is revised and also the new recent references are added (pages 11, 12). OK.

  • -it is necessary to refer more precisely to your results in the discussion, currently, it is presented too generally:

Response: According to the reviewer's comment, the whole discussion section is edited to present the results of the current study and the other studies more precisely (pages 11, 12). OK.

Author Response

Dear Editor-in-Chief of Brain Sciences

Prof. Dr. Stephen D. Meriney

We would like to take this opportunity to thank you and the reviewers for allowing us to improve our manuscript and resubmit it to the Journal of Brain Sciences.

The reviewer's comments have all been addressed in the text and the responses have been highlighted in bold, red font.

Sincerely yours,

Mrs. Mohaddeseh Hafez Yousefi

Response to the comments of the reviewer:

General information:

  • in the abstract, basic information about participants should be added / the results should be more precise

Response: Basic information about participants and results is added in the abstract in lines 18,19,25,26 on page 1.

Reviewer comment: That's not enough. Please add for example: (X women and Y men; mean age: ... ± ... years). The abstract is intended to be the essence of the entire manuscript. It must contain basic information.

New response: Basic information about participants is added in the abstract in lines 18,19 on page 1 and page 5 in table 1 is visible.

  • -first of all, references should be corrected. References are too old

Response: New references (2019, 2022) are added in the introduction and discussion sections of the manuscript.

Reviewer comment: Good, but I still think it is insufficient. Only 3 new articles have been added.

 New response: references 17,18,19,20,21,31,43 and 52 are added some new references in pages 14,15,16.

M&M:

  • in the inclusion criteria: "Suffering from LBP for more than six weeks" but the authors used the term chronic. Chronic pain lasts 12 or more weeks.

Response:  That is right. However, based on the results of some studies, which added in line 100, brain changes towards chronic pain occurred from the 6th week. Accordingly, LBP for more than six weeks is considered chronic pain in the current study.

Reviewer comment: I don't agree. Chronic pain is 12 weeks and over. Additionally, you cite the manuscript.

New response: The main aim of this study was to assess the fear of pain in CLBP from starting the brain changes. Because the brain changes start at 6th week, based on the evidence, we included patients with 6 weeks of pain. However, in agreement with the respected reviewer's comment, chronic LBP is considered from 12 weeks after pain. Therefore, the phrase chronic LBP is replaced with subacute LBP in the whole manuscript.